# Optimization of Filler Compositions of Electrically Conductive Polypropylene Composites for the Manufacturing of Bipolar Plates

**DOI:** 10.3390/polym15143076

**Published:** 2023-07-18

**Authors:** Muhammad Tariq, Nabeel Ahmed Syed, Amir Hossein Behravesh, Remon Pop-Iliev, Ghaus Rizvi

**Affiliations:** Faculty of Engineering and Applied Science, Ontario Tech University, 2000 Simcoe Street North, Oshawa, ON L1G 0C5, Canada; muhammad.tariq7@ontariotechu.net (M.T.); fnu.utkarsh@ontariotechu.net (U.); nabeelahmed.syed@ontariotechu.net (N.A.S.); amir.behravesh@ontariotechu.ca (A.H.B.); remon.pop-iliev@ontariotechu.ca (R.P.-I.)

**Keywords:** MWCNT, carbon black, expanded graphite, optimization, bipolar plates

## Abstract

In this research, polypropylene (PP)–graphite composites were prepared using the melt mixing technique in a twin-screw extruder. Graphite, multi-walled carbon nanotubes (MWCNT), carbon black (CB), and expanded graphite (EG) were added to the PP in binary, ternary, and quaternary formations. The graphite was used as a primary filler, and MWCNT, CB, and EG were added to the PP–graphite composites as secondary fillers at different compositions. The secondary filler compositions were considered the control input factors of the optimization study. A full factorial design of the L-27 Orthogonal Array (OA) was used as a Design of Experiment (DOE). The through-plane electrical conductivity and flexural strength were considered the output responses. The experimental data were interpreted via Analysis of Variance (ANOVA) to evaluate the significance of each secondary filler. Furthermore, statistical modeling was performed using response surface methodology (RSM) to predict the properties of the composites as a function of filler composition. The empirical model for the filler formulation demonstrated an average accuracy of 83.9% and 93.4% for predicting the values of electrical conductivity and flexural strength, respectively. This comprehensive experimental study offers potential guidelines for producing electrically conductive thermoplastic composites for the manufacturing of bipolar fuel cell plates.

## 1. Introduction

A fuel cell converts the chemical energy of the fuel directly into direct current electricity through electrochemical reactions [1]. Fuel cells exhibit higher operational efficiency than internal combustion engines since they directly convert chemical energy into electrical energy without undergoing a combustion reaction and do not contain any moving parts that produce noise during the operation [2]. Additionally, hydrogen fuel cell emits only water and a small amount of heat without producing carbon dioxide. Therefore, a fuel cell is an environmentally friendly, silent, reliable, and fuel-efficient power source [3,4]. Proton exchange membrane (PEM) fuel cells have a relatively low operating temperature and can be used as an eco-friendly power source in various applications, including homes, data centers, portable communication towers, cars, buses, and trains [3,5,6].

The commercialization of fuel cells in transportation and other industries has been obstructed by the fuel cell stack’s weight, cost, and size [3,5,7]. A PEM fuel cell stack is an assembly of multiple PEM fuel cells. Figure 1 presents a schematic diagram of a fuel cell. Each fuel comprises two bipolar plates known as the anode and cathode. A fuel cell stack is formed by connecting a series of bipolar plates with a proton exchange membrane between them [2]. Bipolar plates play a significant role in the operation of PEM fuel cells by distributing hydrogen and oxygen, passing electrons between cells, preventing gases from leaking, and removing the excess heat generated during the electrochemical process [1,8,9]. Bipolar plates, on the other hand, account for 45–60% of a fuel cell’s stack cost, 70–80% of its total weight, and a significant portion of its volume [3,6,10,11]. Due to the high material costs and time-consuming processes involved in fabricating bipolar fuel cell plates, PEM fuel cells are significantly expensive, heavy, and bulky [3,5]. Fuel cell industries have made substantial efforts globally to minimize the material cost, weight, and size of bipolar plates, as well as to find convenient and cost-effective ways to manufacture them [3,6].

A number of comprehensive studies have been published in the literature investigating conductive polymer composites and their applications in various industrial applications [12,13]. There has been increasing interest in electrically conductive composite materials over the past few years, leading researchers across the globe to examine the feasibility of using plastic resin for manufacturing bipolar plates [14,15,16,17]. Using electrically conductive polymer composites for the manufacturing of bipolar plates will reduce the cost, weight, and size and make the production process easier and cost-efficient, helping to commercialize fuel cells in automotive and other industries [11,15,16]. Numerous studies have been conducted to evaluate the viability of thermoset composites for manufacturing bipolar fuel cell plates. The low viscosity of thermoset resin gives an advantage to the manufacturer to manufacture composite at a high filler composition [5]. At higher filler loadings, the thermoset composites usually have better bending strength and toughness than thermoplastic composites [5]. Table 1 summarizes commercially available bipolar plates made from thermoset resins. However, the main concern with thermoset bipolar plates is their low production rate (or slow process) due to the available manufacturing process. Another drawback is that thermosets are not recyclable, which makes them environmentally unfavored. Thermoset plastic cannot be recycled, and due to this limitation, the thermoset bipolar plates cannot be manufactured via mass production methods such as sheet extrusion and injection molding.

The feasibility of producing bipolar plates from thermoplastic composites has been studied extensively in the literature [21,22,23]. It is challenging to manufacture bipolar plates from thermoplastic composites because of their high viscosity, making uniform mixing difficult. The properties of thermoplastic composite bipolar plates are heavily influenced by the type of conductive filler, the total filler content, and the bonding between the filler and the matrix. Increasing the filler content improves electrical conductivity but reduces mechanical performance [5]. Thus, one of the most important challenging issues is to enhance electrical properties while maintaining the mechanical properties in acceptable ranges [24]. Hence, filler contents must be optimized to achieve optimal electrical conductivity and flexural strength for bipolar plates [3]. Adequate through-plane electrical conductivity is required for electrons to travel from one cell to another in a fuel cell pack. The electrical conductivity of the composite material in the through-plane direction is many times lower than that in the in-plane direction [25,26].

Lawrence R.J. [27] tested graphite-based PVDF composites at 86 wt.% of graphite content and recorded the electrical conductivity of 277 S/cm in the in-plane direction. Herrera et al. [28] investigated the mechanical properties of polypropylene composites and found considerable improvement in flexural strength and tensile strength on adding MWCNT and carbon nanofibers to the polypropylene matrix. Yan et al. [29] mixed carbon nanotubes with polypropylene at different compositions and recorded the electrical conductivity of the composites as a function of CNT content. Alo et al. [26] used a polypropylene/epoxy blend as the binding matrix for the composites and recorded the value of in-plane and through-plane electrical conductivity as a function of total filler content. They observed that the in-plane and through-plane electrical conductivities at 50 wt.% filler contents were 49.26 S/cm and 0.37 S/cm, respectively, while at 85 wt.% filler content, these results were 90.34 S/cm and 9.34 S/cm, respectively. Dhakate et al. [30] investigated the effects of EG on the in-plane electrical conductivity of the composites. Adloo et al. [31] recorded the in-plane electrical conductivity value of 14.92 S/cm and the bending strength of 50.9 MPa by adding 6 wt.% of CB and 66 wt.% of graphite to the PP matrix. Liao et al. [32] recorded the in-plane electrical conductivity of 420.6 S/cm with a flexural strength of 19.6 MPa by adding 22.5 wt.% graphite and 2.5 wt.% graphite to the polyethylene. Lee et al. [33] developed PP/CB composites for electrically conductive polymer composite coating to protect aluminum bipolar plates from corrosion.

Selamat et al. [34] conducted an optimization study on compression molding parameters to obtain optimum electrical conductivity and flexural strength of polypropylene composites. Roncaglia et al. [35] used a two-level, full-factorial design to optimize pressure, mold temperature, and time for the manufacturing of graphite–epoxy composites to maximize electrical conductivity. Fatma et al. [36] analyzed the effects of the surface contact angle and surface roughness of composite bipolar plates on the performance of PEM fuel cell performance using the response surface methodology. King et al. [15] conducted an optimization study for the filler content by adding carbon black, graphite, and carbon nanotubes to the polypropylene at different levels. They observed the maximum through-plane electrical conductivity of 38.31 S/cm at the combination of 6 wt.% carbon nanotubes, 2.5 wt.% CB, and 65 wt.% of synthetic graphite. Many well-documented research studies have focused on developing electrically conductive composites by adding different fillers to polymer matrixes. Graphite, carbon black, carbon nanotubes, and expanded graphite have been used as conductive fillers to develop electrically conductive polymer composites. However, neither a comprehensive study investigating the synergistic effects of these fillers in binary, ternary, and quaternary filler configurations, nor an optimization study to determine the filler content that would optimize the electrical conductivity and flexural strength, has been well documented in the literature.

The effects of total filler content on the electrical and mechanical properties of the PP/composites and the interaction of binary fillers with the graphite were investigated in the previous study by plotting the properties of the composites as a function of total filler content [37]. In continuation of the previous study, the total filler content was fixed at 75 wt.% in the current research work, and PP composites were produced by changing the compositions of primary and secondary fillers. Graphite was used as a primary filler, whereas MWCNT, CB, and EG were added as secondary fillers. These fillers were mixed with the PP in binary, ternary, and quaternary filler formations to investigate the interactions between each filler. The parametric evaluation of the secondary filler compositions has been carried out to achieve the optimum values of electrical conductivity and flexural strength. A full-factorial design approach was used to implement the DOE. The significance of each secondary filler was evaluated using ANOVA. A mathematical model was also developed by using RSM to predict the values of electrical conductivity and flexural strength of the composites.

## 2. Experimental

### 2.1. Materials

Homopolymer Polypropylene PP-3620WZ, with a mass flow rate of 12 g/10 min, was received from Total Petrochemicals USA Inc., Houstan, TX, USA. The primary graphite filler used in this study was prepared by mixing synthetic graphite (Asbury Carbons^®^ A99) with natural graphite (Asbury Carbons^®^ GP3243) in weight proportions of 10% and 90%. The same configuration of graphite filler was used in a previous study [37]. Carbon black Ketjenblack EC-600JD, with a surface area of 1400 m^2^/g, was received from Akzo Noble^®^. A PP-MWCNT masterbatch CNT-PP-25, with the MWCNT weight composition of 25%, was purchased from CTI Materials Inc. The expandable graphite flake (#808121) was obtained from Sigma-Aldrich^®^, St. Louis, MO, USA. The expandable graphite was expanded in the lab furnace at 700 °C for a period of 15 min.

### 2.2. Experimental Set-Up

A Leistritz co-rotating twin-screw extruder (ZSE18), with a screw L/D ratio of 40, was used for the melt compounding of fillers with the polymer matrix. The extruder has two feeders and eight heating zones. The polypropylene pellets were inserted in Zone 1 through the main feeder, while graphite was introduced in Zone 4 through the side feeder. The MWCNT was added from the main feeder in the form of a PP-MWCNT masterbatch. The CB was dry-mixed with graphite and supplied through the side feeder. In the case of EG composites, a masterbatch of PP and EG was prepared in the first step and sent to the main feeder in the second step.

The materials were mixed in the extruder, and a composite material strand with a diameter of 3 mm was extruding out from the die orifice. The composite strand was cooled down in the water bath and then pelletized using the pelletizer (Figure 2a). After compounding, the pelletized composite material was oven-dried to remove the moisture. The composite material was processed by compression molding inside a Carver press to produce the disc-shaped specimen (Figure 2b). A five-inch diameter disc with a thickness of 1.5 mm has been produced for each composite. The processing temperature was 190 °C. The pressure was set to 1000 psi for all types of composites. At the beginning of the process, the press was preheated to the processing temperature, and then the mold was placed inside the preheated press. The heat was provided for 15 min to maintain the temperature, and then the mold was set for cooling, whereas the pressure of 1000 psi was maintained until the mold temperature was cooled down to 100 °C.

The final composition of the composite material for each run was validated using the Thermo-Gravimetric Analyzer, TGA Q50, by TA Instruments, USA. The TGA thermo-gram of experimental trail #1 is shown in Figure 3. The weight of the composite material decreased as the temperature increased, and this can be attributed to the degradation of the polymer. Once the temperature exceeded 500 °C, the polymer component of the composite was completely degraded, resulting in a residue that clearly represented the filler content of the composite. The TGA analysis validated that the actual filler composition of the composite was 74.93 wt.%, which closely matched the intended composition.

## 3. Characterization

### 3.1. Through-Plane Electrical Conductivity (TPEC)

The composite materials were tested for TPEC in accordance with the guidelines of the US Fuel Cell Council through-plane electrical conductivity testing protocol [38]. The specimen was compressed between two gold-plated copper electrodes. Gas diffusion layers, Toray carbon paper TGP-H120, were placed between the specimen and electrodes to improve contact. The details of the testing process are described elsewhere [37]. A total of five samples for each filler composition were cut by maintaining the sample size of 1 by 1 inches.

### 3.2. Flexural Strength Testing

A 3-point bending test was performed to determine the flexural strength of the composites. A Dynamic Mechanical Analyzer, DMA Q800 by TA instruments, was used to perform flexural strength testing. All the specimens had a width of 12.7 mm and a thickness of 1.5 mm. The support span of the 3-point bending clamp was 50 mm. ASTM D790 protocol was used to calculate the crosshead speed. A total of five samples of each composition were tested, and the average value was recorded.

### 3.3. Scanning Electron Microscopy (SEM)

A scanning electron microscope was used to investigate the morphological characteristics of the composite samples. In SEM analysis, an electron microscope is used to produce images by scanning the surface of the specimen with the help of an electron beam. The SEM images of the composite material were produced using a Hitachi FlexSEM 1000. The samples were gold-coated and placed on conductive copper tape for better image processing. The operating voltage was 20 kV. The magnification of the microscope was adjusted according to the types of composites.

### 3.4. Design of Experiments

The input factors of the DOE and their levels are mentioned in Table 2. Each factor has three levels of variations. The total filler content was fixed at 75 wt.% for each run, wherein the composition of graphite was configured according to the compositions of the secondary fillers. In the open literature, it was observed that the addition of a small amount of carbon nanotubes substantially improves the electrical conductivity of the composites. The addition of carbon nanotubes above 4 wt.% does not significantly affect the electrical conductivity [39,40]. Hence, 4 wt.% was set as the maximum level of MWCNT composition. The maximum amount of CB used in this study was 5 wt.%. The same composition of CB as a secondary filler has been used in multiple research studies [41,42,43]. The researchers observed a sudden jump in the electrical conductivity of EG composites when the EG content was between 10 wt.% to 30 wt.% [30,44,45]. Based on these findings, the upper limit for the EG content was set to 30 wt.% in the experimental design. A full-factorial design based on three control factors with three-level variation was designed to investigate the effect of each secondary filler and the possibility of any interactions between the fillers. The design is based on 27 experimental runs with two output responses. ANOVA was used to study the significance of each input parameter. Table 3 represents the values of the process parameters and the output response for the DOE.

## 4. Results and Discussion

### 4.1. Effects of Filler Interaction on the Electrical Conductivity

Five specimens of each experimental trial were tested. The average values of the measured electrical conductivity and the standard deviation are given in Table 3 for all samples. High electrical conductivity values were observed in Trials #15, 21, and 27, wherein high levels of CB and MWCNT were present. The highest electrical conductivity was recorded at trial #27 using 4 wt.% MWCNT, 30 wt.% EG, and 5 wt.% CB.

The effects of secondary filler on the conductivity network inside the composite material can be observed in the captured SEM micrographs, as given in Figure 4. The electrical conductivity of single filler PP/graphite composite was 2.8 S/cm for Trial 1. The presence of a polymer layer between the graphite particles can be detected in Figure 4a. This layer acts as insulation between graphite particles, preventing electron hopping, which results in a low electrical conductivity. The introduction of MWCNT in PP/graphite composites demonstrated promising performance in terms of electrical conductivity. The addition of 4 wt.% MWCNT to PP/graphite composite in Trial #19 resulted in an increase in electrical conductivity by more than seven times, reaching 20.5 S/cm. The high aspect ratio of MWCNT allows the creation of connections between graphite particles, thus forming new conductive paths for the electrons, which lowers the electrical resistivity. Figure 4b shows the conductive network formed by MWCNT. Similar effects on the electrical conductivity of composites using carbon nanotubes are reported in previous studies. King et al. [15] reported that the electrical conductivity was increased from 0.29 S/cm to 17.9 S/cm upon adding 6 wt.% of carbon nanotubes to the PP/graphite composites. Pötschke et al. [39] added MWCNT to polycarbonate and observed a tenfold increase in electrical conductivity of the composites as the MWCNT load was increased from 1 wt.% to 1.5 wt.%.

A significant improvement in the electrical conductivity was observed upon adding CB as a binary filler to the PP/graphite composites. The addition of 5 wt.% CB increased the electrical conductivity from 2.8 S/cm to 19.3 S/cm. The CB particles create links between the graphite particles due to their smaller particle size and higher surface area, thereby forming additional electrical paths and improving electrical conductivity. Figure 4d illustrates the conductive network formed by the presence of CB particles. Numerous research studies have reported a significant enhancement in the electrical properties when CB is used as a binary filler. Kang et al. [41] investigated the effects of CB as a binary filler in the graphite–phenolic-resin composites for fuel cell applications and observed a considerable increase in the electrical conductivity on adding 5 wt.% of CB. King et al. [15] observed the reduction in the electrical resistivity of PP/graphite composites from 3.43 to 0.36 Ω cm with the addition of 2.5 wt.% CB. Heiser et al. [42] found that the electrical conductivity of nylon/graphite composites was significantly improved by adding 5 wt.% CB.

Incorporating EG in the PP/graphite composites positively affected electrical conductivity. Adding 30 wt.% of EG to the PP/graphite composites increased the electrical conductivity from 2.8 S/cm to 4.6 S/cm. However, the increase observed in this study was not significant because the effectiveness of adding EG particles to the electrical conductivity mainly depends on the conductive linkages formed inside the composites. The vermicular shape, porous structure, and high aspect ratio of EG particles promote the formation of continuous conductive paths within the polymer matrix [46,47]. The mixing technique is crucial in preserving the particle structure from potential disintegration during the manufacturing of composites [48]. Like natural graphite particles, EG has a layered structure but a broader spacing between the layers [49]. The loose structure of EG makes it a soft and porous material [30]. The deterioration of EG particles during the processing might affect its ability to form electrical connections. The high shear force generated during the melt-compounding inside the twin-screw extruder could cause EG particles to break apart. The fragmentation of EG particles can be seen in Figure 4c. Similar studies on the manufacturing of EG composites at low rpm inside the internal mixture or using solution processing resulted in better electrical properties. Dhakate et al. [30] added EG to the phenolic resin and studied the effect of EG on the electrical conductivity (in-plane) of the composites. The EG particles were dry-mixed with Novolac phenolic resin powder and then processed in a compression mold. The electrical conductivity of 110 S/cm was achieved in the in-plane direction at 40 wt.% of EG content; they did not report through-plane conductivity. Wu et al. [44] mixed EG particles with PP in an internal mixer at 60 rpm for 10 min. They observed a sudden jump in the electrical conductivity when the EG content was increased from 10 to 15 wt.%. Sever et al. [45] added EG to high-density polyethylene. The mixing was carried out in an internal mixer at the mixing speed of 35 rpm for 15 min, and the electrical conductivity was dramatically increased with 10 wt.% EG.

The interactions between the secondary fillers in ternary and quaternary filler configuration can be seen in Figure 4e–i. The CB was added as a ternary filler to the PP/graphite/MWCNT composites and demonstrated significant improvement in the electrical conductivity. The electrical conductivity of the PP/graphite/MWCNT/CB composite, Trial #21, is almost equal to the sum of the electrical conductivity of PP/graphite/CB and PP/graphite/MWCNT composites in Trials #3 and #19, respectively. The addition of CB in PP/graphite/MWCNT composites substantially enhanced the conductivity network formed by MWCNT particles. The conductive linkages made by the combination of CB and MWCNT can be seen in Figure 4g. King et al. [15] used 2.5 wt.% of CB as a ternary filler in PP/graphite/MWCNT composites and observed an increase in electrical conductivity from 17.9 to 37.3 S/cm. As a ternary filler, CB also demonstrated promising results when added to the PP/graphite/EG composites. The fragments of EG particles, when combined with the CB, formed electrical paths between graphite particles, which can be seen in Figure 4f. The maximum value of electrical conductivity observed in PP/graphite/EG/CB composites was 24.6 S/cm. The conductivity values of PP/graphite/EG/CB composites were significantly higher than those of PP/graphite/EG composites, but no significant difference was found compared with PP/graphite/CB composites, indicating triviality of EG. The addition of EG as a ternary filler to the PP/graphite/MWCNT composites exhibited adverse effects on the electrical conductivity. The conductivity of PP/graphite/MWCNT composites decreased from 20.5 S/cm to 12.1 S/cm with 30 wt.% of EG. This could be attributed to the deformation of EG particles during the melt-mixing process. The high shear stress generated during the melt-compounding process caused the rigid particles of MWCNT to penetrate through the soft and porous EG particles. The penetration of MWCNT inside the EG can be detected in Figure 4e. Therefore, the EG particles break the conductive links established by MWCNT rather than creating additional electrical paths. The interaction between MWCNT and EG particles was improved with the addition of CB particles. The interaction of graphite, MWCNT, EG, and CB within the PP/graphite/MWCNT/EG/CB composites can be seen in Figure 4h,i. The combination of MWCNT, EG, and CB particles led to the formation of a complex conductive network that interlinked the graphite particles.

### 4.2. Effects of Filler Interaction on the Flexural Strength

The flexural strength of five specimens from each composition was tested, and the average values and the standard deviation are given in Table 3. Experimental runs involving 2.5 wt.% CB and 0 wt.% MWCNT demonstrated high flexural strength values. The highest flexural strength attained in this study was 36.9 MPa at Trial #2 with 0 wt.% MWCNT, 0 wt.% EG, and 2.5 wt.% CB. In contrast with the strength measurement, the electrical conductivity of this composition was 9.7 S/cm.

The mechanical performance of the composites was improved by adding a small amount of CB. However, incorporating CB at a high content demonstrated adverse effects on flexural strength. The flexural strength of PP/graphite composites increased from 30.3 to 36.9 MPa when 2.5 wt.% CB was added but decreased to 34.2 MPa when CB was added up to 5 wt.%. A similar trend was observed when CB was added in ternary and quaternary filler formations. The high surface area of CB results in better interfacial bonding with the resin, leading to better mechanical properties. On the other hand, the high content of CB inside the composite absorbs a large amount of resin, causing poor wetting of graphite and reducing the overall mechanical properties [41]. This phenomenon was also observed in a research work investigated by Kang et al. [41], wherein the feasibility of using CB was studied to develop a lightweight fuel cell stack. In the study, CB was added as a binary filler in the graphite–phenolic-resin composites up to 5 wt.%. The flexural strength was observed to increase with the addition of 1 wt.% CB but started to decrease with the increase in CB content to more than 3 wt.%.

The MWCNT exhibited adverse effects on the mechanical properties of the composites. The overall flexural strength of the composites was observed to decrease with the increase in MWCNT content. It may be attributed to improper filler mixing, causing the MWCNT particles to agglomerate. The melt-compounding technique is not very effective in handling the agglomeration issues of MWCNT [50,51]. Dhakate et al. [52] observed that adding 2 vol.% of MWCNT to polymer/graphite composites reduced the bending strength. The incorporation of EG has not been observed with any noticeable effects on the flexural strength of the composites. The flexural strength slightly decreased with increasing EG content in PP/graphite and PP/graphite/CB composites. This may be due to the weak interfacial bonding of the porous structure of EG with the resin [46]. However, a slight improvement in the flexural strength was observed when EG was mixed with MWCNT composites. The variation in the values of flexural strength on changing EG content was not significant.

### 4.3. Analysis of Variance

The parametric evaluation of the experimental design was performed using ANOVA software. The control factors were evaluated, and their significance was investigated. The mean output response for three control factors at each level is mentioned in Table 4. The values A, B, and C represent the summation values of MWCNT, EG, and CB, respectively. The term SS is the sum of the squared deviations, and the value V is the mean square (variance). The F-value is a ratio of the mean square (V) to the mean square error. The *p*-value is the probability of obtaining an F-ratio. It was observed that the compositions of MWCNT (A) and CB (C) demonstrated significant effects on the electrical conductivity and flexural strength of the composite. The F-test values for MWCNT and CB in terms of electrical conductivity were found to be 53.1 and 98.6, respectively, with a corresponding *p*-value of 0.0001. These results clearly indicate the high significance of MWCNT and CB on electrical conductivity at a 99% confidence level, with a lower risk level. Similarly, in the case of flexural strength, the F-test values for MWCNT and CB were 10.1 and 5.846, respectively, also demonstrating the high significance of these factors on flexural strength at a confidence level of 99%. In contrast, the composition of the EG showed the most negligible impact on the conductivity and strength measurements. The F-values of EG were observed to be 0.48 and 1.22 for electrical conductivity and flexural strength measurements, respectively, which were much lower than the F value (2.59) at a confidence level of 90%. Based on its minimum statistical summation of electrical conductivity and flexural strength, EG was comparatively deemed insignificant.

### 4.4. Factorial Design Analysis

#### 4.4.1. Electrical Conductivity

The summation values of the electrical conductivity for each input factor are shown in Figure 5a. It was observed that the variations in CB and MWCNT content demonstrated a significant impact on the electrical conductivity that supports the previous findings. In contrast, the electrical conductivity summation values for EG did not show any significant fluctuations. It was found that increasing the composition level of MWCNT from A_1_ to A_3_ (0 wt.% to 4 wt.%) and CB from C_1_ to C_3_ (0 wt.% to 5 wt.%) significantly increased the value of electrical conductivity. Moreover, the difference between the lowest and the highest summation values of electrical conductivity for the MWCNT and CB were found to be 138.8 and 190 S/cm, respectively, whereas this difference was only 13.3 S/cm in the case of EG. These results confirm that the composition of CB is the most significant factor, with the highest variability effects on the electrical conductivity, whereas EG has the least effect on the electrical conductivity.

#### 4.4.2. Flexural Strength

Figure 5b represents the summation values of flexural strength for each input factor. The summation values for the MWCNT contents demonstrated the highest variation, whereas the EG contents showed the least variation. Summation values for CB were high when the composition level of CB increased from C_1_ to C_2_ (0 wt.% to 2.5 wt.%) but declined when the composition increased to C_3_ (5 wt.%). The CB particles contribute to establishing strong interfacial bonding with the resin due to their large surface area, resulting in improved flexural strength. At the same time, the high surface area of the filler particle also interfaces with a large amount of polymeric material. Thus, the increased content of CB promotes contact with a significant volume of the resin, leaving less of the polymer for graphite wetting, leading to inefficient mixing and poor interfacial bonding between graphite and the polymer matrix [41]. The analysis showed that with the increase in the CB composition from C_1_ to C_2_ (0 wt.% to 2.5 wt.%) and the decrease in MWCNT composition from A_3_ to A_1_ (4 wt.% to 0 wt.%), the flexural strength increased and reached the highest value. The difference between the highest and the lowest summation values for MWCNT, CB, and EG was 46.2, 13.14, and 35.46 MPa, respectively. These results confirm that the CB content is the most significant factor influencing electrical conductivity, while the EG content has the least significant effect on electrical conductivity.

### 4.5. Response Surface Methodology

In this study, the RSM was employed as a statistical regression tool to empirically analyze and establish the relationship between input variables and output response. RSM is a practical modeling technique that utilizes polynomial regressions to capture the intricate relationship between the output response and input variables rather than relying on approximations. The main objective of implementing RSM was to predict the output response, which was influenced by three independent variables.

The statistical study investigates the effect of three input factors, viz., MWCNT, EG, and CB, with a three-level variation. A full-factorial OA was chosen for this study, comprising all possible factor combinations, which lead to 3^3^ = 27 trials for the complete analysis. By employing the full-factorial OA, the investigator can assess both the main effects and the interactions of the input variables. This design enables a thorough evaluation of how the factors individually and collectively influence the outcome. Multiple regression equations were generated to evaluate the significance of control factors on the electrical conductivity and flexural strength as output responses. Mathematical modeling was performed on these output responses using the RSM technique to predict the values of electrical conductivity and flexural strength as a function of filler compositions. The empirical model generated for the composite manufacturing process is given by the following equations:(1)$$X=1.4296+(5.2444\times A)-(0.1204\times B)+(3.5733\times C)-(0.3472\times A^{2})+(0.0049\times B^{2})+(0.1298\times C^{2})$$
(2)$$Y=29.6756-(2.3261\times A)-(0.022\times B)+(2.7951\times C)+(0.2558\times A^{2})+(0.0027\times B^{2})-(0.4699\times C^{2})$$
where X is the predicted value of electrical conductivity (S/cm) and Y is the predicted value of flexural strength (MPa), A is the composition of MWCNT (wt.%), B is the composition of EG (wt.%), and C is the composition of CB (wt.%).

Figure 6a,b shows that the average accuracy differentiates between the experimental data and the predictive output response. The predictive model demonstrated an average model accuracy of 84% and 94% for the electrical conductivity (X) and flexural strength (Y), respectively, based on 27 experimental runs. The three-dimensional surface contour plots of the output responses are demonstrated in Figure 7a,b.

### 4.6. Comparison of Experimental Results with the Commercially Available Materials

The properties of the thermoplastic composites developed in this research were compared with those of commercially available thermoset bipolar plates listed in Table 1. BMC-940-15252A, produced by A. Schulman (LyondellBasell, Fairlawn, OH, USA), has TPEC value of 25 S/cm, with a flexural strength of 56 MPa. The flexural strength values of the thermoplastic composites listed in Table 3 are comparatively lower than those of the thermoset bipolar plates. However, the flexural strength values of most of the formulations are above the criterion of the US Department of Energy, which is 25 MPa. Thermoset composites are stronger than thermoplastics, but their primary disadvantage is their low production rate. Thermoplastic composites bipolar plates can be produced through high production-rate processes such as injection molding and sheet extrusion. Experimental run #27 exhibits the most optimal properties in terms of electrical conductivity and flexural strength. It achieved a TPEC value of 39.6 S/cm, which surpasses the TPEC values of the commercially available materials mentioned in Table 1. Notably, it represents the highest TPEC value recorded in this experimental study. The flexural strength value of this formulation is 29.4 MPa, meeting the criteria set by the US Department of Energy. However, due to the incorporation of MWCNT, the material cost is relatively higher than that of other formulations. In future studies, there is potential to explore ways to reduce the overall material cost. Equations (1) and (2) derived from the RSM can be integrated with the cost of materials to facilitate cost optimization efforts. By considering both the performance characteristics and material costs, a more economically viable composite formulation can be achieved.

## 5. Conclusions

In this experimental study, electrically conductive PP composites were produced using a melt-compounding technique in a twin-screw extruder by adding selected conductive fillers of MWCNT, EG, and CB. Conductive fillers were added to the polymer in binary, ternary, and quaternary formulations to synergistically enhance the electrical conductivity and flexural strength of the composites. The effects of filler contents at three levels of variation were studied using a full-factorial design. The composition with 4 wt.% MWCNT, 5 wt.% CB, and 30 wt.% EG demonstrated the optimum values of the response parameters with the through-plane electrical conductivity of 39.6 S/cm and flexural strength of 29.4 MPa. The ANOVA study validates the significance of MWCNT and CB compositions at the high confidence level of 99% with a low risk level. The RSM-generated mathematical model exhibits an average accuracy of 83.9% and 93.4% for electrical conductivity and flexural strength, respectively, for the range of selected process parameters. The experimental results present a promising approach in the open literature to thoroughly understand the synergistic effects of conductive fillers on the electrical and mechanical properties of the electrically conductive thermoplastic composites.

## Figures and Tables

**Figure 1 polymers-15-03076-f001:**
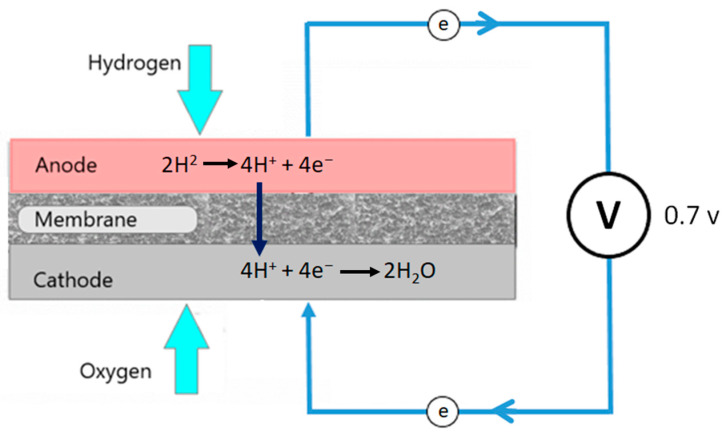
Schematic diagram of PEM fuel cell.

**Figure 2 polymers-15-03076-f002:**
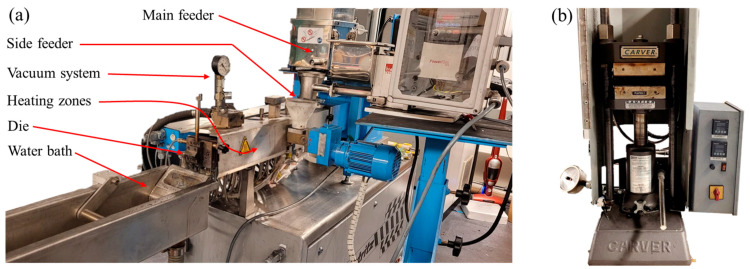
Experimental set-up: (**a**) twin-screw extruder; (**b**) compression molding machine.

**Figure 3 polymers-15-03076-f003:**
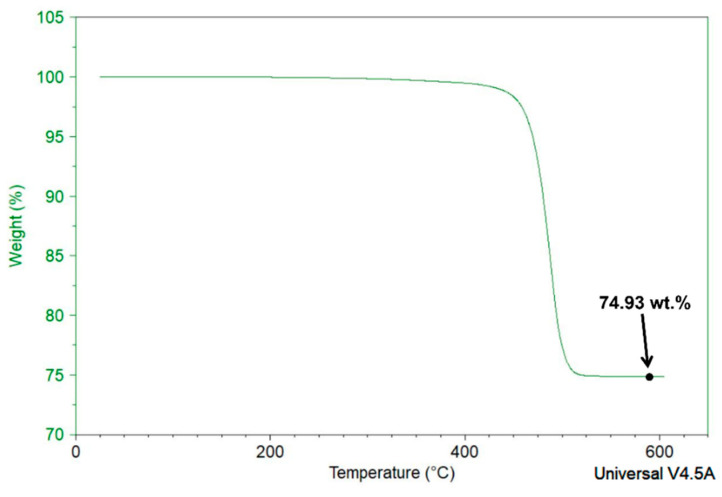
TGA analysis for 75 wt.% filled composite material.

**Figure 4 polymers-15-03076-f004:**
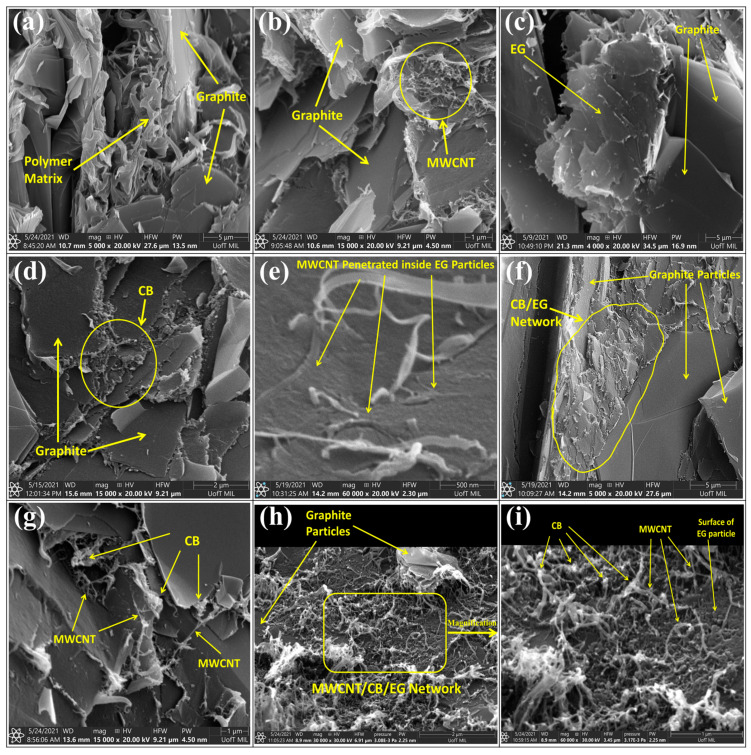
SEM micrographs of PP composites: (**a**) PP/graphite, (**b**) PP/graphite/MWCNT, (**c**) PP/graphite/EG, (**d**) PP/graphite/CB, (**e**) PP/graphite/MWCNT/EG, (**f**) PP/graphite/EG/CB, (**g**) P/graphite/MWCNT/CB, (**h**) PP/graphite/MWCNT/EG/CB, and (**i**) magnified view of PP/graphite/MWCNT/EG/CB.

**Figure 5 polymers-15-03076-f005:**
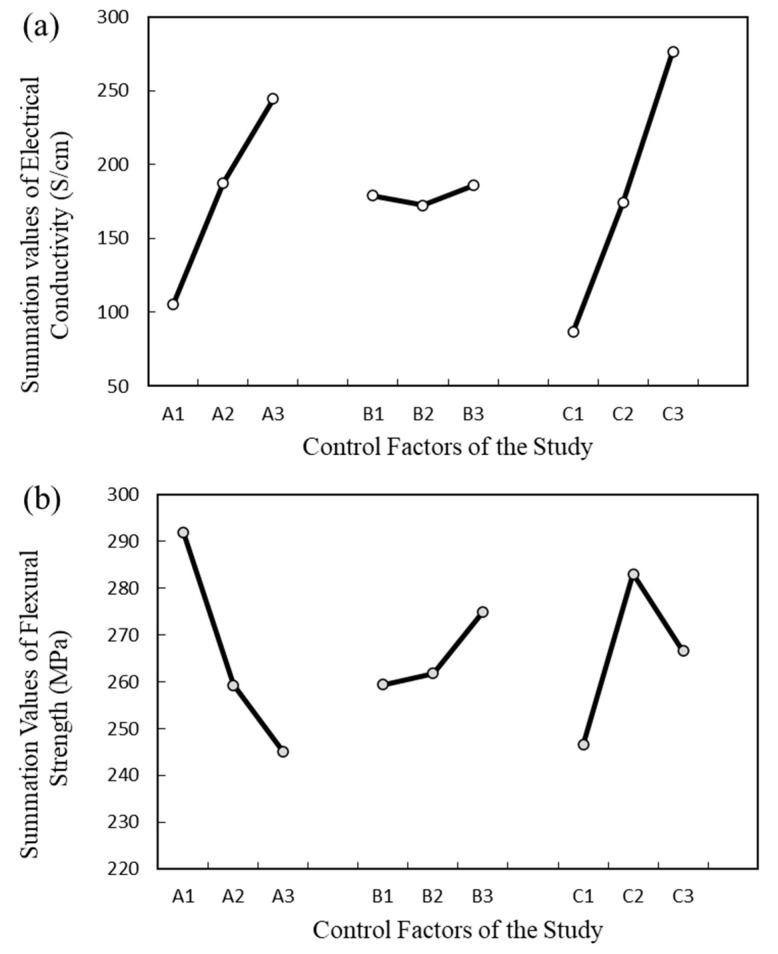
Summation values of the output responses: (**a**) electrical conductivity; (**b**) flexural strength.

**Figure 6 polymers-15-03076-f006:**
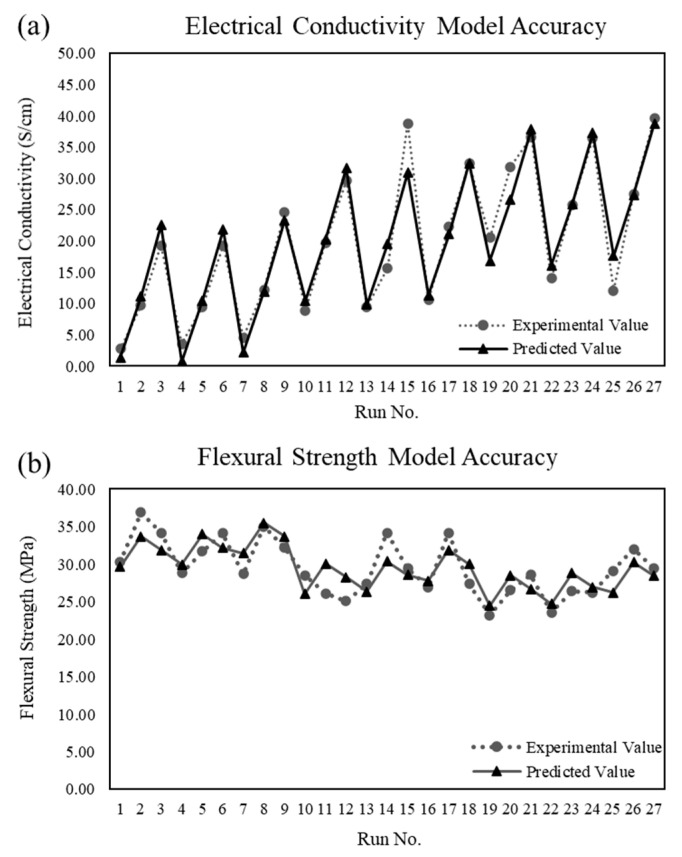
Comparison of predictive model outcomes with experimental outcomes: (**a**) electrical conductivity; (**b**) flexural strength.

**Figure 7 polymers-15-03076-f007:**
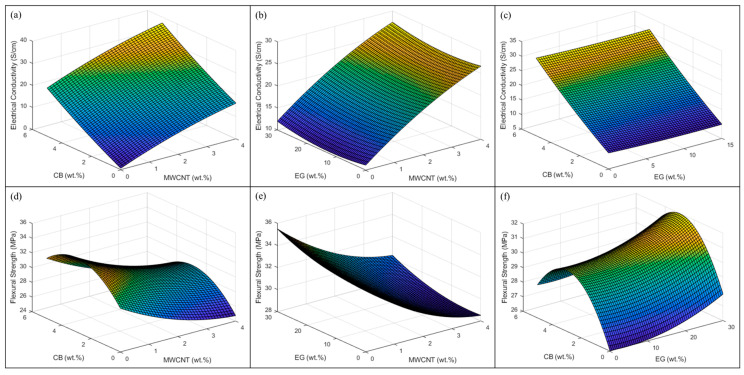
Three-dimensional surface contour plots for the following measurements: electrical conductivity (**a**–**c**) and flexural strength (**d**–**f**).

**Table 1 polymers-15-03076-t001:** List of thermoset composite bipolar plates.

Manufacturers/Patents	Binding Matrix	Graphite Content (wt.%)	Electrical Conductivity (S/cm)		Flexural Strength (MPa)
			In-Plane	Through-Plane	
DuPont [5]	-	-	-	25–33	53
GTI (US) [18]	Phenolic	77.5	53	-	-
BMC940-15252A [19]	Vinyl Ester	-	133	25	56
SGL [5]	-	-	100	20	40
Plug Power [5]	Vinyl Ester	68	55	-	40
ORNL (US) [20]	Phenolic	Carbon Fiber	200	-	-

**Table 2 polymers-15-03076-t002:** Control factors and levels of the DOE.

Control Factors	Symbol	Level 1	Level 2	Level 3
MWCNT (wt.%)	A	0	2	4
EG (wt.%)	B	0	15	30
CB (wt.%)	C	0	2.5	5

**Table 3 polymers-15-03076-t003:** Output responses of DOE.

Run#	A	B	C	Electrical Conductivity (S/cm) (Through-Plane)	Flexural Strength (MPa)
1	0	0	0	2.8 ± 0.2	30.3 ± 1.3
2	0	0	2.5	9.7 ± 0.4	36.9 ± 1.2
3	0	0	5	19.3 ± 1.2	34.2 ± 2.8
4	0	15	0	3.6 ± 0.1	28.8 ± 0.4
5	0	15	2.5	9.5 ± 0.2	31.7 ± 2.3
6	0	15	5	19.3 ± 1.1	34.1 ± 2.3
7	0	30	0	4.6 ± 0.1	28.7 ± 1.4
8	0	30	2.5	12.2 ± 0.5	35.0 ± 2.0
9	0	30	5	24.6 ± 1.5	32.2 ± 2.8
10	2	0	0	8.9 ± 0.2	28.5 ± 2.1
11	2	0	2.5	19.7 ± 0.5	26.1 ± 3.1
12	2	0	5	29.6 ± 0.7	25.1 ± 0.9
13	2	15	0	9.5 ± 0.2	27.4 ± 2.6
14	2	15	2.5	15.7 ± 0.4	34.2 ± 3.8
15	2	15	5	38.8 ± 0.9	29.4 ± 1.8
16	2	30	0	10.6 ± 0.3	26.9 ± 4.4
17	2	30	2.5	22.3 ± 1.1	34.2 ± 1.9
18	2	30	5	32.4 ± 0.3	27.4 ± 0.6
19	4	0	0	20.5 ± 1.3	23.2 ± 1.9
20	4	0	2.5	31.9 ± 0.8	26.5 ± 2.0
21	4	0	5	36.6 ± 0.6	28.6 ± 3.2
22	4	15	0	14.0 ± 0.2	23.6 ± 1.8
23	4	15	2.5	25.8 ± 0.6	26.4 ± 2.9
24	4	15	5	36.4 ± 0.2	26.2 ± 2.6
25	4	30	0	12.1 ± 0.6	29.1 ± 2.7
26	4	30	2.5	27.5 ± 0.4	32.0 ± 3.8
27	4	30	5	39.6 ± 1.1	29.4 ± 1.7

**Table 4 polymers-15-03076-t004:** Analysis of Variance results.

Analysis of Variance (ANOVA)								
Electrical Conductivity	A1	105.6	B1	179	C1	86.6	(SS)Total	3304.939
	A2	187.5	B2	172.6	C2	174.3		
	A3	244.4	B3	185.9	C3	276.6		
	SSA	1081.876	SSB	9.831852	SSC	2009.503	(SS)Error	203.7274
	VA	540.9381	VB	4.915926	VC	1004.751	(V)Error	10.18637
	(F)A	53.10411	(F)B	0.482598	(F)C	98.63685		
	(p)A	0.0001	(p)B	0.6242	(p)C	0.0001		
Flexural Strength	A1	291.9	B1	259.4	C1	246.54	(SS)Total	344.8688
	A2	259.24	B2	261.8	C2	283		
	A3	245	B3	274.94	C3	266.6		
	SSA	128.4838	SSB	15.55227	SSC	74.09982	(SS)Error	126.7329
	VA	64.24191	VB	7.776133	VC	37.04991	(V)Error	6.336644
	(F)A	10.13816	(F)B	1.227169	(F)C	5.846929		
	(p)A	0.0009	(p)B	0.3143	(p)C	0.01		
Significant @ 90% confidence level							2.59	
Significant @ 95% confidence level							3.49	
Significant @ 99% confidence level							5.85	

## Data Availability

The data used to support the findings of this study are included within the article.

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
