# Peer review of "Optimization of Filler Compositions of Electrically Conductive Polypropylene Composites for the Manufacturing of Bipolar Plates"

_polymers, 2023, doi:10.3390/polym15143076_

Round 1

Reviewer 1 Report

In this manuscript, the authors use MWCNT, carbon black, and expanded graphite as fillers in polypropylene, trying to optimize the filler ratio for electrical conductivity as well as mechanical performance for conductive polymer composites in fuel cells. Simulation works were further performed. The experiment design can be improved to be more scientific. Overall, the work is complete with an acceptable level of novelty. I would suggest acceptance after major revision.

11.      Can the authors discuss the motivation of using PP as the base material instead of other polymer materials?

22.      Why 4%, 30% and 5% are set as the upper limit ratio for the fillers? The authors may need to discuss the experiment design.

33.      Since the best performance is achieved by the 4% MWCNT, 30% EG and 5% CB, which are all the highest adding ratio of all three groups of fillers, does that means the authors have not yet found the optimized ratio for this combination? It is also related to the 2nd question. I would like to see their new experiment results with further increased additive ratio. 

44.      The incorporation of EG seems to have minor improvements on both electrical conductivity and mechanical performance, but the adding ratio is up to 30%, which is quite a lot as a filler. Then what is the meaning of adding it? Have the authors observed any drawbacks of adding such large amount of EG into PP, e.g. dispersion issues, agglomerations?  Have the authors consider to exclude EG and further increase the ratio of CNT and CB?

Author Response

We would like to express our sincere appreciation to the reviewer for their valuable comments and suggestions. The feedback provided by the reviewer has greatly contributed to the overall improvement of the manuscript.

Comments to the Author

In this manuscript, the authors use MWCNT, carbon black, and expanded graphite as fillers in polypropylene, trying to optimize the filler ratio for electrical conductivity as well as mechanical performance for conductive polymer composites in fuel cells. Simulation works were further performed. The experiment design can be improved to be more scientific. Overall, the work is complete with an acceptable level of novelty. I would suggest acceptance after major revision.

  1. Can the authors discuss the motivation of using PP as the base material instead of other polymer materials?

A number of research studies on electrically conductive thermoplastic composites found in the literature are based on Polypropylene. These studies are referenced in Section 1 (Introduction) of the manuscript. Moreover, the current research builds upon a previous study (https://doi.org/10.1002/er.7898 ) in which conductive fillers were incorporated into PP at varying filler levels, ranging from 0 wt.% to 85 wt.%.

  1. Why 4%, 30% and 5% are set as the upper limit ratio for the fillers? The authors may need to discuss the experiment design.

The reason for selecting the upper limits of secondary fillers has been discussed in Section 3.4.

  1. Since the best performance is achieved by the 4% MWCNT, 30% EG and 5% CB, which are all the highest adding ratio of all three groups of fillers, does that means the authors have not yet found the optimized ratio for this combination? It is also related to the 2nd question. I would like to see their new experiment results with further increased additive ratio.

The research data mentioned in this article is a part of an ongoing industrial project. The brief description of the project can be found on the following link,

https://www.mitacs.ca/en/projects/material-formulation-and-sheet-extrusion-thermoplastic-graphite-composites-compression

The purpose of this project is to develop an electrically conductive thermoplastic composites that can be used for the production of fuel cell bipolar plates. The research team is trying to develop a formulation that has electrical conductivity and flexural strength values of above 25 S/cm and 25 MPa, respectively. A couple of material formulation presented in Table 4 have surpassed these values.Therefore, there is no need to further increase the additive compositions. However, these numbers are confidential and can not be disclosed in the article.

  1. The incorporation of EG seems to have minor improvements on both electrical conductivity and mechanical performance, but the adding ratio is up to 30%, which is quite a lot as a filler. Then what is the meaning of adding it? Have the authors observed any drawbacks of adding such large amount of EG into PP, e.g. dispersion issues, agglomerations? Have the authors consider to exclude EG and further increase the ratio of CNT and CB?

The decision to incorporate EG into the composites was based on findings from various studies in the literature, which reported significant improvements in electrical conductivity with the use of EG. Consequently, the research team had anticipated favorable results by adding EG to the composites. However, contrary to expectations, the addition of EG did not lead to the desired improvement. Notably, no issues related to dispersion or agglomeration were observed. The primary reason for the ineffectiveness of EG in this study is attributed to the fragmentation of EG particles during the melt-compounding process. The high shear forces generated within the twin-screw extruder during melt-compounding can cause the breakdown of EG particles. These factors have been thoroughly discussed in Section 4.1 of the manuscript. As a result, the authors have decided to exclude EG from the composite formulation in the future studies.

Reviewer 2 Report

The work presented by M. Tariq et al. analyses and optimises the polypropylene-carbon based composited for bipolar plates by experimental data and statistical model. The introduction is well written, and it points out the importance of the filler on conductivity and mechanical properties of composites.

Couple of remarks

The authors mention in the abstract “A full factorial design of the L-27 Orthogonal Array (OA) “ . Couple of words should be added to explain the term or to be added to line 384.

Can the TGA curves be added in the manuscript or in the supporting info rmation?

Can the authors develop a little more the section 4.5 Response Surface Methodology? The RSM can predict the properties of the composites as a function of other types and percentages of filler composition.

Not all the references are completed, most of them are missing the DOI and the patents are missing the number

Author Response

First of all, the authors would like to thank the reviewer for the raised comments, which significantly enhanced the quality of the current work. A detailed action for each comment is provided below. All changes have been highlighted with yellow color in the revised manuscript.

Comments to the Author

       1.      The authors mention in the abstract “A full factorial design of the L-27 Orthogonal Array (OA) “ . Couple of words should be added to explain the term or to be added to line 384.

The description for the full factorial orthogonal array has been incorporated in Section 4.5 at Line 423. Additionally, Line 384 in the previous version of the manuscript has been updated to Line 423.

  1.     Can the TGA curves be added in the manuscript or in the supporting information?

The TGA thermo-gram of one of the experimental trails has been added (Figure 2) to the manuscript, and the text in Section 2.2 has been modified accordingly. The changes in the revised manuscript have been highlighted in yellow.

  1.   Can the authors develop a little more the section 4.5 Response Surface Methodology? The RSM can predict the properties of the composites as a function of other types and percentages of filler composition.

We have taken your suggestion into account and have enhanced the content in Section 4.5, providing additional details on Response Surface Methodology (RSM).

  1.     Not all the references are completed, most of them are missing the DOI and the patents are missing the number

The authors are thankful to the reviewer for highlighting this issue. The DOI and patent numbers have been added to the references.

Reviewer 3 Report

1.            In section 1. Introduction, it is recommended to add a sketch of a PEM fuel cell with Bipolar plates.

2.            In section 1. Introduction, it is recommended to describe the optimal values for electrical and mechanical properties for composite bipolar plates.

3.            In section 2.2. Experimental Set-up, it is recommended to add a diameter of composite material strand. And the pulling speed of composite material strand.

4.            In section 2.2. Experimental Set-up, it is recommended to add the technological mode of compression molding. Add dimensions of disc-shaped specimen. It is recommended to add photos of strand and disc-shaped specimens.

5.            In section 3.1. Through-Plane Electrical Conductivity (TPEC), it is recommended to add sample sizes.

6.            In section 3.2. Flexural Strength Testing, it is recommended to add sample sizes.

7.            In section 3. Characterization, it is recommended to add a description of the method for obtaining SEM micrographs of PP composites.

8.            In section 4.3. Analysis of Variation, it is recommended to describe the results in more detail in Table 4. Decipher all the abbreviations in Table 4 in the text.

9.            In section 4.5. Response Surface Methodology, in equations (1) and (2), specify the dimensions of all terms.

10.          Since the title contains the word “optimization”, it is necessary to show the result of optimization. Which trial became optimal and why?

11.          It is necessary to discuss the significance of the results obtained. To make a comparison with the results obtained earlier. Give recommendations on optimizing the content of fillers. To discuss the possibility of using the recommended electrically conductive PP composites for serial production of bipolar plates.

12.          How are you planning to continue this research?

Author Response

The authors would like to thank the reviewer for the raised comments, which significantly enhanced the quality of the current work. A detailed action for each comment is provided below. All changes have been highlighted with yellow color in the revised manuscript.

Comments to the Author

1.      In section 1. Introduction, it is recommended to add a sketch of a PEM fuel cell with Bipolar plates.

A schematic diagram of PEM fuel cell has been added in section 1. The changes in Section 1 have been highlighted in yellow.

 2      - In section 1. Introduction, it is recommended to describe the optimal values for electrical and mechanical properties for composite bipolar plates.

It is difficult to describe the optimal values for electrical conductivity and flexural strength. The research data mentioned in this paper is a part of an ongoing industrial project. The brief description of the project can be found on the following link,

https://www.mitacs.ca/en/projects/material-formulation-and-sheet-extrusion-thermoplastic-graphite-composites-compression

The bipolar plates manufactured by the industry are made up of thermoset composites. The industry currently utilizes a thermoset composite material, and the main goal of this project is to develop an electrically conductive thermoplastic-based composite material that can be used to produce fuel cell bipolar plates. However, thermosets composites are stronger and tougher than thermoplastic composites; therefore, it is challenging to achieve these properties in thermoplastic composites. The research team is diligently working to maximize the flexural strength while maintaining an electrical conductivity above 25 S/cm in through-plane direction. Unfortunately, the electrical conductivity and flexural strength values of the thermoset composite material that the industry is currently using are confidential and can not be disclosed in the paper.  However, for comparison purposes, the electrical conductivity and flexural strength values of the commercially available composite bipolar plates are mentioned in Table 1.

  1.     In section 2.2. Experimental Set-up, it is recommended to add a diameter of composite material strand. And the pulling speed of composite material strand.

The diameter of the composite strand was 3 mm. The diameter of the strand has been added to the manuscript (line 159). However, the pulling speed of the strand was different in each experimental run. The speed was directly influenced by the material flow rate, which was typically around 12 grams per minute, causing the strand to be extruded at a speed of approximately 1.65 cm/s. However, achieving an exact flow rate with the main feeder posed challenges, making it difficult to maintain a consistent speed. Consequently, the flow rate of the side feeder was modified to correspond to the main feeder's flow rate, resulting in varying material flow rates and strand speeds across different experimental runs.

  1.     In section 2.2. Experimental Set-up, it is recommended to add the technological mode of compression molding. Add dimensions of disc-shaped specimen. It is recommended to add photos of strand and disc-shaped specimens.

The details of the disc-shaped specimen along with operating parameters for the compression molding have been added in Section 2.2. The photos of twin-screw extruder and compression molding machine were already provided in Figure 2. However, the photos of the strand and disc-shaped specimen taken during the experimentation are inconclusive, and the quality of the images is not sufficient for publication purposes.

  1.    In section 3.1. Through-Plane Electrical Conductivity (TPEC), it is recommended to add sample sizes.

The sample size for the electrical conductivity testing has been added in Section 3.1

  1.    In section 3.2. Flexural Strength Testing, it is recommended to add sample sizes.

The sample dimensions for the flexural strength testing have been mentioned in Section 3.2.

  1.     In section 3. Characterization, it is recommended to add a description of the method for obtaining SEM micrographs of PP composites

The authors are thankful to the reviewer for highlighting the shortcomings in the manuscript. A new section, Section 3.3, has been included in the manuscript, providing detailed information about the SEM method.

  1.     In section 4.3. Analysis of Variation, it is recommended to describe the results in more detail in Table 4. Decipher all the abbreviations in Table 4 in the text.

We have taken your suggestion into consideration. The results presented in Table 4 have been further elaborated in Section 4.3. Additionally, we have ensured that all abbreviations used in Table 4 are clarified and defined in the corresponding text.

  1.    In section 4.5. Response Surface Methodology, in equations (1) and (2), specify the dimensions of all terms.

The dimensions of all the terms of equation (1) and (2) have been described.

  1.     Since the title contains the word “optimization”, it is necessary to show the result of optimization. Which trial became optimal and why?

We have taken it into account, and in response, we have included the details of the optimal trial in Section 4.6 of the manuscript.

  1.     It is necessary to discuss the significance of the results obtained. To make a comparison with the results obtained earlier. Give recommendations on optimizing the content of fillers. To discuss the possibility of using the recommended electrically conductive PP composites for serial production of bipolar plates.

We sincerely appreciate the reviewer's valuable suggestions. In response to these comments, we have included a new section in the manuscript, Section 4.6. This section extensively discusses the significance of the results obtained in our experimental study and compares them with the properties of commercially available products. We believe these additions significantly enhance the discussion and provide valuable insights into both the current findings and the future potential of our research.

  1. How are you planning to continue this research?

The research can be extended by incorporation of toughness modifiers into the polymer resin to further enhance the mechanical properties of the composites.

Round 2

Reviewer 1 Report

For the first time in my 10 years' research, a revised manuscript without a point by point response is presented in front of me. The authors completely ignored my comments/discussions. I personally don't think it acceptable. I will not give further comments and leave it to the editors. 

Round 3

Reviewer 1 Report

Can be accepted in present form.